# WASSERSTEIN AUTO-ENCODERS

**Ilya Tolstikhin**
MPI for Intelligent Systems
Tübingen, Germany
`ilya@tue.mpg.de`

**Olivier Bousquet**
Google Brain
Zürich, Switzerland
`obousquet@google.com`

**Sylvain Gelly**
Google Brain
Zürich, Switzerland
`sylvaingelly@google.com`

**Bernhard Schölkopf**
MPI for Intelligent Systems
Tübingen, Germany
`bs@tue.mpg.de`

## ABSTRACT

We propose the Wasserstein Auto-Encoder (WAE)—a new algorithm for building a generative model of the data distribution. WAE minimizes a penalized form of the Wasserstein distance between the model distribution and the target distribution, which leads to a different regularizer than the one used by the Variational Auto-Encoder (VAE) (Kingma & Welling, 2014). This regularizer encourages the encoded training distribution to match the prior. We compare our algorithm with several other techniques and show that it is a generalization of adversarial auto-encoders (AAE) (Makhzani et al., 2016). Our experiments show that WAE shares many of the properties of VAEs (stable training, encoder-decoder architecture, nice latent manifold structure) while generating samples of better quality, as measured by the FID score.

## 1 INTRODUCTION

The field of representation learning was initially driven by supervised approaches, with impressive results using large labelled datasets. Unsupervised generative modeling, in contrast, used to be a domain governed by probabilistic approaches focusing on low-dimensional data. Recent years have seen a convergence of those two approaches. In the new field that formed at the intersection, variational auto-encoders (VAEs) (Kingma & Welling, 2014) constitute one well-established approach, theoretically elegant yet with the drawback that they tend to generate blurry samples when applied to natural images. In contrast, generative adversarial networks (GANs) (Goodfellow et al., 2014) turned out to be more impressive in terms of the visual quality of images sampled from the model, but come without an encoder, have been reported harder to train, and suffer from the "mode collapse" problem where the resulting model is unable to capture all the variability in the true data distribution. There has been a flurry of activity in assaying numerous configurations of GANs as well as combinations of VAEs and GANs. A unifying framework combining the best of GANs and VAEs in a principled way is yet to be discovered.

This work builds up on the theoretical analysis presented in Bousquet et al. (2017). Following Arjovsky et al. (2017); Bousquet et al. (2017), we approach generative modeling from the optimal transport (OT) point of view. The OT cost (Villani, 2003) is a way to measure a distance between probability distributions and provides a much weaker topology than many others, including $f$-divergences associated with the original GAN algorithms (Nowozin et al., 2016). This is particularly important in applications, where data is usually supported on low dimensional manifolds in the input space $\mathcal{X}$. As a result, stronger notions of distances (such as $f$-divergences, which capture the density ratio between distributions) often max out, providing no useful gradients for training. In contrast, OT was claimed to have a nicer behaviour (Arjovsky et al., 2017; Gulrajani et al., 2017) although it requires, in its GAN-like implementation, the addition of a constraint or a regularization term into the objective.

Figure 1: Both VAE and WAE minimize two terms: the reconstruction cost and the regularizer penalizing discrepancy between $P_Z$ and distribution induced by the encoder $Q$. VAE forces $Q(Z|X = x)$ to match $P_Z$ for all the different input examples $x$ drawn from $P_X$. This is illustrated on picture (a), where every single red ball is forced to match $P_Z$ depicted as the white shape. Red balls start intersecting, which leads to problems with reconstruction. In contrast, WAE forces the continuous mixture $Q_Z := \int Q(Z|X)dP_X$ to match $P_Z$, as depicted with the green ball in picture (b). As a result latent codes of different examples get a chance to stay far away from each other, promoting a better reconstruction.

In this work we aim at minimizing OT $W_c(P_X, P_G)$ between the true (but unknown) data distribution $P_X$ and a *latent variable model $P_G$* specified by the prior distribution $P_Z$ of latent codes $Z \in \mathcal{Z}$ and the generative model $P_G(X|Z)$ of the data points $X \in \mathcal{X}$ given $Z$. Our main contributions are listed below (cf. also Figure 1):

- A new family of regularized auto-encoders (Algorithms 1, 2 and Eq. 4), which we call *Wasserstein Auto-Encoders* (WAE), that minimize the optimal transport $W_c(P_X, P_G)$ for any cost function $c$. Similarly to VAE, the objective of WAE is composed of two terms: the $c$-reconstruction cost and a regularizer $\mathcal{D}_Z(P_Z, Q_Z)$ penalizing a discrepancy between two distributions in $\mathcal{Z}$: $P_Z$ and a distribution of encoded data points, i.e. $Q_Z := \mathbb{E}_{P_X}[Q(Z|X)]$. When $c$ is the squared cost and $\mathcal{D}_Z$ is the GAN objective, WAE coincides with adversarial auto-encoders of Makhzani et al. (2016).

- Empirical evaluation of WAE on MNIST and CelebA datasets with squared cost $c(x, y) = \|x - y\|_2^2$. Our experiments show that WAE keeps the good properties of VAEs (stable training, encoder-decoder architecture, and a nice latent manifold structure) while generating samples of *better quality*, approaching those of GANs.

- We propose and examine two different regularizers $\mathcal{D}_Z(P_Z, Q_Z)$. One is based on GANs and adversarial training *in the latent space $\mathcal{Z}$*. The other uses the maximum mean discrepancy, which is known to perform well when matching high-dimensional standard normal distributions $P_Z$ (Gretton et al., 2012). Importantly, the second option leads to a fully adversary-free min-min optimization problem.

- Finally, the theoretical considerations presented in Bousquet et al. (2017) and used here to derive the WAE objective might be interesting in their own right. In particular, Theorem 1 shows that in the case of generative models, *the primal form* of $W_c(P_X, P_G)$ is equivalent to a problem involving the optimization of a probabilistic encoder $Q(Z|X)$.

The paper is structured as follows. In Section 2 we review a novel auto-encoder formulation for OT between $P_X$ and the latent variable model $P_G$ derived in Bousquet et al. (2017). Relaxing the resulting constrained optimization problem we arrive at an objective of Wasserstein auto-encoders. We propose two different regularizers, leading to WAE-GAN and WAE-MMD algorithms. Section 3 discusses the related work. We present the experimental results in Section 4 and conclude by pointing out some promising directions for future work.

## 2 PROPOSED METHOD

Our new method minimizes the optimal transport cost $W_c(P_X, P_G)$ based on the novel auto-encoder formulation (see Theorem 1 below). In the resulting optimization problem the decoder tries to accurately reconstruct the encoded training examples as measured by the cost function $c$. The encoder tries to simultaneously achieve two conflicting goals: it tries to match the encoded distribution of training examples $Q_Z := \mathbb{E}_{P_X}[Q(Z|X)]$ to the prior $P_Z$ as measured by any specified divergence $\mathcal{D}_Z(Q_Z, P_Z)$, while making sure that the latent codes provided to the decoder are informative enough to reconstruct the encoded training examples. This is schematically depicted on Fig. 1.

### 2.1 PRELIMINARIES AND NOTATIONS

We use calligraphic letters (i.e. $\mathcal{X}$) for sets, capital letters (i.e. $X$) for random variables, and lower case letters (i.e. $x$) for their values. We denote probability distributions with capital letters (i.e. $P(X)$) and corresponding densities with lower case letters (i.e. $p(x)$). In this work we will consider several measures of discrepancy between probability distributions $P_X$ and $P_G$. The class of *f-divergences* (Liese & Miescke, 2008) is defined by $D_f(P_X \| P_G) := \int f\left(\frac{p_X(x)}{p_G(x)}\right) p_G(x)dx$, where $f: (0, \infty) \to \mathcal{R}$ is any convex function satisfying $f(1) = 0$. Classical examples include the Kullback-Leibler $D_{\mathrm{KL}}$ and Jensen-Shannon $D_{\mathrm{JS}}$ divergences.

### 2.2 OPTIMAL TRANSPORT AND ITS DUAL FORMULATIONS

A rich class of divergences between probability distributions is induced by the *optimal transport* (OT) problem (Villani, 2003). Kantorovich's formulation of the problem is given by

$$W_c(P_X, P_G) := \inf_{\Gamma \in \mathcal{P}(X \sim P_X, Y \sim P_G)} \mathbb{E}_{(X,Y) \sim \Gamma}[c(X, Y)], \tag{1}$$

where $c(x, y) \colon \mathcal{X} \times \mathcal{X} \to \mathcal{R}_+$ is any measurable *cost function* and $\mathcal{P}(X \sim P_X, Y \sim P_G)$ is a set of all joint distributions of $(X, Y)$ with marginals $P_X$ and $P_G$ respectively. A particularly interesting case is when $(\mathcal{X}, d)$ is a metric space and $c(x, y) = d^p(x, y)$ for $p \geq 1$. In this case $W_p$, the $p$-th root of $W_c$, is called *the $p$-Wasserstein distance*.

When $c(x, y) = d(x, y)$ the following Kantorovich-Rubinstein duality holds[1]:

$$W_1(P_X, P_G) = \sup_{f \in \mathcal{F}_L} \mathbb{E}_{X \sim P_X}[f(X)] - \mathbb{E}_{Y \sim P_G}[f(Y)], \tag{2}$$

where $\mathcal{F}_L$ is the class of all bounded 1-Lipschitz functions on $(\mathcal{X}, d)$.

### 2.3 APPLICATION TO GENERATIVE MODELS: WASSERSTEIN AUTO-ENCODERS

One way to look at modern generative models like VAEs and GANs is to postulate that they are trying to minimize certain discrepancy measures between the data distribution $P_X$ and the model $P_G$. Unfortunately, most of the standard divergences known in the literature, including those listed above, are hard or even impossible to compute, especially when $P_X$ is unknown and $P_G$ is parametrized by deep neural networks. Previous research provides several tricks to address this issue.

In case of minimizing the KL-divergence $D_{\mathrm{KL}}(P_X, P_G)$, or equivalently maximizing the marginal log-likelihood $E_{P_X}[\log p_G(X)]$, the famous *variational lower bound* provides a theoretically grounded framework successfully employed by VAEs (Kingma & Welling, 2014; Mescheder et al., 2017). More generally, if the goal is to minimize the $f$-divergence $D_f(P_X, P_G)$ (with one example being $D_{\mathrm{KL}}$), one can resort to its dual formulation and make use of $f$-GANs and *the adversarial training* (Nowozin et al., 2016). Finally, OT cost $W_c(P_X, P_G)$ is yet another option, which can be, thanks to the celebrated Kantorovich-Rubinstein duality (2), expressed as an adversarial objective as implemented by the Wasserstein-GAN (Arjovsky et al., 2017). We include an extended review of all these methods in Supplementary A.

---

[1]Note that the same symbol is used for $W_p$ and $W_c$, but only $p$ is a number and thus the above $W_1$ refers to the 1-Wasserstein distance.

In this work we will focus on *latent variable models* $P_G$ defined by a two-step procedure, where first a code $Z$ is sampled from a fixed distribution $P_Z$ on a latent space $\mathcal{Z}$ and then $Z$ is mapped to the image $X \in \mathcal{X} = \mathcal{R}^d$ with a (possibly random) transformation. This results in a density of the form

$$p_G(x) := \int_{\mathcal{Z}} p_G(x|z) p_z(z) dz, \quad \forall x \in \mathcal{X}, \tag{3}$$

assuming all involved densities are properly defined. For simplicity we will focus on non-random decoders, i.e. generative models $P_G(X|Z)$ deterministically mapping $Z$ to $X = G(Z)$ for a given map $G \colon \mathcal{Z} \to \mathcal{X}$. Similar results for random decoders can be found in Supplementary B.1.

It turns out that under this model, the OT cost takes a simpler form as the transportation plan factors through the map $G$: instead of finding a coupling $\Gamma$ in (1) between two random variables living in the $\mathcal{X}$ space, one distributed according to $P_X$ and the other one according to $P_G$, it is sufficient to find a conditional distribution $Q(Z|X)$ such that its $Z$ marginal $Q_Z(Z) := \mathbb{E}_{X \sim P_X}[Q(Z|X)]$ is identical to the prior distribution $P_Z$. This is the content of the theorem below proved in Bousquet et al. (2017). To make this paper self contained we repeat the proof in Supplementary B.

**Theorem 1** *For $P_G$ as defined above with deterministic $P_G(X|Z)$ and any function $G \colon \mathcal{Z} \to \mathcal{X}$*

$$\inf_{\Gamma \in \mathcal{P}(X \sim P_X, Y \sim P_G)} \mathbb{E}_{(X,Y) \sim \Gamma}\big[c(X,Y)\big] = \inf_{Q \colon Q_Z = P_Z} \mathbb{E}_{P_X} \mathbb{E}_{Q(Z|X)}\big[c(X, G(Z))\big],$$

*where $Q_Z$ is the marginal distribution of $Z$ when $X \sim P_X$ and $Z \sim Q(Z|X)$.*

This result allows us to optimize over random encoders $Q(Z|X)$ instead of optimizing over all couplings between $X$ and $Y$. Of course, both problems are still constrained. In order to implement a numerical solution we relax the constraints on $Q_Z$ by adding a penalty to the objective. This finally leads us to the WAE objective:

$$D_{\mathrm{WAE}}(P_X, P_G) := \inf_{Q(Z|X) \in \mathcal{Q}} \mathbb{E}_{P_X} \mathbb{E}_{Q(Z|X)}\big[c(X, G(Z))\big] + \lambda \cdot \mathcal{D}_Z(Q_Z, P_Z), \tag{4}$$

where $\mathcal{Q}$ is any nonparametric set of probabilistic encoders, $\mathcal{D}_Z$ is an arbitrary divergence between $Q_Z$ and $P_Z$, and $\lambda > 0$ is a hyperparameter. Similarly to VAE, we propose to use deep neural networks to parametrize both encoders $Q$ and decoders $G$. Note that as opposed to VAEs, the WAE formulation allows for non-random encoders deterministically mapping inputs to their latent codes.

We propose two different penalties $\mathcal{D}_Z(Q_Z, P_Z)$:

**GAN-based $\mathcal{D}_Z$.** The first option is to choose $\mathcal{D}_Z(Q_Z, P_Z) = D_{\mathrm{JS}}(Q_Z, P_Z)$ and use the adversarial training to estimate it. Specifically, we introduce an adversary (discriminator) in the latent space $\mathcal{Z}$ trying to separate[2] "true" points sampled from $P_Z$ and "fake" ones sampled from $Q_Z$ (Goodfellow et al., 2014). This results in the WAE-GAN described in Algorithm 1. Even though WAE-GAN falls back to the min-max problem, we move the adversary from the input (pixel) space $\mathcal{X}$ to the latent space $\mathcal{Z}$. On top of that, $P_Z$ may have a nice shape with a single mode (for a Gaussian prior), in which case the task should be easier than matching an unknown, complex, and possibly multi-modal distributions as usually done in GANs. This is also a reason for our second penalty:

**MMD-based $\mathcal{D}_Z$.** For a positive-definite reproducing kernel $k \colon \mathcal{Z} \times \mathcal{Z} \to \mathcal{R}$ the following expression is called *the maximum mean discrepancy* (MMD):

$$\mathrm{MMD}_k(P_Z, Q_Z) = \Big\| \int_{\mathcal{Z}} k(z, \cdot) dP_Z(z) - \int_{\mathcal{Z}} k(z, \cdot) dQ_Z(z) \Big\|_{\mathcal{H}_k},$$

where $\mathcal{H}_k$ is the RKHS of real-valued functions mapping $\mathcal{Z}$ to $\mathcal{R}$. If $k$ is *characteristic* then $\mathrm{MMD}_k$ defines a *metric* and can be used as a divergence measure. We propose to use $\mathcal{D}_Z(P_Z, Q_Z) = \mathrm{MMD}_k(P_Z, Q_Z)$. Fortunately, MMD has an unbiased U-statistic estimator, which can be used in conjunction with stochastic gradient descent (SGD) methods. This results in the WAE-MMD described in Algorithm 2. It is well known that the maximum mean discrepancy performs well when matching high-dimensional standard normal distributions (Gretton et al., 2012) so we expect this penalty to work especially well working with the Gaussian prior $P_Z$.

---

[2]We noticed that the famous "log trick" (also called "non saturating loss") proposed by Goodfellow et al. (2014) leads to better results.

---

**ALGORITHM 1** Wasserstein Auto-Encoder with GAN-based penalty (WAE-GAN).

**Require:** Regularization coefficient $\lambda > 0$.
  Initialize the parameters of the encoder $Q_\phi$, decoder $G_\theta$, and latent discriminator $D_\gamma$.
  **while** $(\phi, \theta)$ not converged **do**
    Sample $\{x_1, \ldots, x_n\}$ from the training set
    Sample $\{z_1, \ldots, z_n\}$ from the prior $P_Z$
    Sample $\tilde{z}_i$ from $Q_\phi(Z|x_i)$ for $i = 1, \ldots, n$
    Update $D_\gamma$ by ascending:

$$\frac{\lambda}{n} \sum_{i=1}^{n} \log D_\gamma(z_i) + \log\big(1 - D_\gamma(\tilde{z}_i)\big)$$

    Update $Q_\phi$ and $G_\theta$ by descending:

$$\frac{1}{n} \sum_{i=1}^{n} c\big(x_i, G_\theta(\tilde{z}_i)\big) - \lambda \cdot \log D_\gamma(\tilde{z}_i)$$

**end while**

---

**ALGORITHM 2** Wasserstein Auto-Encoder with MMD-based penalty (WAE-MMD).

**Require:** Regularization coefficient $\lambda > 0$, characteristic positive-definite kernel $k$.
  Initialize the parameters of the encoder $Q_\phi$, decoder $G_\theta$, and latent discriminator $D_\gamma$.
  **while** $(\phi, \theta)$ not converged **do**
    Sample $\{x_1, \ldots, x_n\}$ from the training set
    Sample $\{z_1, \ldots, z_n\}$ from the prior $P_Z$
    Sample $\tilde{z}_i$ from $Q_\phi(Z|x_i)$ for $i = 1, \ldots, n$
    Update $Q_\phi$ and $G_\theta$ by descending:

$$\frac{1}{n} \sum_{i=1}^{n} c\big(x_i, G_\theta(\tilde{z}_i)\big) + \frac{\lambda}{n(n-1)} \sum_{\ell \neq j} k(z_\ell, z_j)$$

$$+ \frac{\lambda}{n(n-1)} \sum_{\ell \neq j} k(\tilde{z}_\ell, \tilde{z}_j) - \frac{2\lambda}{n^2} \sum_{\ell, j} k(z_\ell, \tilde{z}_j)$$

**end while**

---

We point out once again that the encoders $Q_\phi(Z|x)$ in Algorithms 1 and 2 can be non-random, i.e. deterministically mapping input points to the latent codes. In this case $Q_\phi(Z|x) = \delta_{\mu_\phi(x)}$ for a function $\mu_\phi \colon \mathcal{X} \to \mathcal{Z}$ and in order to *sample* $\tilde{z}_i$ from $Q_\phi(Z|x_i)$ we just need to return $\mu_\phi(x_i)$.

## 3   RELATED WORK

**Literature on auto-encoders** Classical unregularized auto-encoders minimize only the reconstruction cost. This results in different training points being encoded into non-overlapping zones chaotically scattered all across the $\mathcal{Z}$ space with "holes" in between where the decoder mapping $P_G(X|Z)$ has never been trained. Overall, the encoder $Q(Z|X)$ trained in this way does not provide a useful representation and sampling from the latent space $\mathcal{Z}$ becomes hard (Bengio et al., 2013).

Variational auto-encoders (Kingma & Welling, 2014) minimize a variational bound on the KL-divergence $D_{\mathrm{KL}}(P_X, P_G)$ which is composed of the reconstruction cost plus the regularizer $\mathbb{E}_{P_X}\big[D_{\mathrm{KL}}(Q(Z|X), P_Z)\big]$. The regularizer captures how distinct the image by the encoder of *each* training example is from the prior $P_Z$, which is not guaranteeing that the overall encoded distribution $\mathbb{E}_{P_X}[Q(Z|X)]$ matches $P_Z$ like WAE does. Also, VAEs require non-degenerate (i.e. non-deterministic) Gaussian encoders and random decoders for which the term $\log p_G(x|z)$ can be computed and differentiated with respect to the parameters. Later Mescheder et al. (2017) proposed a way to use VAE with non-Gaussian encoders. WAE minimizes the optimal transport $W_c(P_X, P_G)$ and allows both probabilistic and deterministic encoder-decoder pairs of any kind.

The VAE regularizer can be also equivalently written (Hoffman & Johnson, 2016) as a sum of $D_{\mathrm{KL}}(Q_Z, P_Z)$ and a mutual information $\mathbb{I}_Q(X, Z)$ between the images $X$ and latent codes $Z$ jointly distributed according to $P_X \times Q(Z|X)$. This observation provides another intuitive way to explain a difference between our algorithm and VAEs: WAEs simply drop the mutual information term $\mathbb{I}_Q(X, Z)$ in the VAE regularizer.

When used with $c(x, y) = \|x - y\|_2^2$ WAE-GAN is equivalent to adversarial auto-encoders (AAE) proposed by Makhzani et al. (2016). Theory of Bousquet et al. (2017) (and in particular Theorem 1) thus suggests that AAEs minimize the 2-Wasserstein distance between $P_X$ and $P_G$. This provides the first theoretical justification for AAEs known to the authors. WAE generalizes AAE in two ways: first, it can use any cost function $c$ in the input space $\mathcal{X}$; second, it can use any discrepancy measure $\mathcal{D}_Z$ in the latent space $\mathcal{Z}$ (for instance MMD), not necessarily the adversarial one of WAE-GAN.

Finally, Zhao et al. (2017b) independently proposed a regularized auto-encoder objective similar to Bousquet et al. (2017) and our (4) based on very different motivations and arguments. Following

VAEs their objective (called InfoVAE) defines the reconstruction cost in the image space *implicitly* through the negative log likelihood term $-\log p_G(x|z)$, which should be properly normalized for all $z \in \mathcal{Z}$. In theory VAE and InfoVAE can both induce arbitrary cost functions, however in practice this may require an estimation of the normalizing constant (partition function) which can[3] be different for different values of $z$. WAEs specify the cost $c(x, y)$ *explicitly* and don't constrain it in any way.

**Literature on OT** Genevay et al. (2016) address computing the OT cost in large scale using SGD and sampling. They approach this task either through the dual formulation, or via a regularized version of the primal. They do not discuss any implications for generative modeling. Our approach is based on the primal form of OT, we arrive at regularizers which are very different, and our main focus is on generative modeling.

The WGAN (Arjovsky et al., 2017) minimizes the 1-Wasserstein distance $W_1(P_X, P_G)$ for generative modeling. The authors approach this task from the dual form. Their algorithm comes without an encoder and can not be readily applied to any other cost $W_c$, because the neat form of the Kantorovich-Rubinstein duality (2) holds only for $W_1$. WAE approaches the same problem from the primal form, can be applied for any cost function $c$, and comes naturally with an encoder.

In order to compute the values (1) or (2) of OT we need to handle non-trivial constraints, either on the coupling distribution $\Gamma$ or on the function $f$ being considered. Various approaches have been proposed in the literature to circumvent this difficulty. For $W_1$ Arjovsky et al. (2017) tried to implement the constraint in the dual formulation (2) by clipping the weights of the neural network $f$. Later Gulrajani et al. (2017) proposed to relax the same constraint by penalizing the objective of (2) with a term $\lambda \cdot \mathbb{E}\left(\|\nabla f(X)\| - 1\right)^2$ which should not be greater than 1 if $f \in \mathcal{F}_L$. In a more general OT setting of $W_c$ Cuturi (2013) proposed to penalize the objective of (1) with the KL-divergence $\lambda \cdot D_{\mathrm{KL}}(\Gamma, P \otimes Q)$ between the coupling distribution and the product of marginals. Genevay et al. (2016) showed that this entropic regularization drops the constraints on functions in the dual formulation as opposed to (2). Finally, in the context of *unbalanced optimal transport* it has been proposed to relax the constraint in (1) by regularizing the objective with $\lambda \cdot \left(D_f(\Gamma_X, P) + D_f(\Gamma_Y, Q)\right)$ (Chizat et al., 2015; Liero et al., 2015), where $\Gamma_X$ and $\Gamma_Y$ are marginals of $\Gamma$. In this paper we propose to relax OT in a way similar to the unbalanced optimal transport, i.e. by adding additional divergences to the objective. However, we show that in the particular context of generative modeling, only one extra divergence is necessary.

**Literature on GANs** Many of the GAN variations (including $f$-GAN and WGAN) come without an encoder. Often it may be desirable to reconstruct the latent codes and use the learned manifold, in which cases these models are not applicable.

There have been many other approaches trying to blend the adversarial training of GANs with auto-encoder architectures (Zhao et al., 2017a; Dumoulin et al., 2017; Ulyanov et al., 2017; Berthelot et al., 2017). The approach proposed by Ulyanov et al. (2017) is perhaps the most relevant to our work. The authors use the discrepancy between $Q_Z$ and the distribution $\mathbb{E}_{Z' \sim P_Z}[Q(Z|G(Z'))]$ of auto-encoded noise vectors as the objective for the max-min game between the encoder and decoder respectively. While the authors showed that the saddle points correspond to $P_X = P_G$, they admit that encoders and decoders trained in this way have no incentive to be reciprocal. As a workaround they propose to include an additional reconstruction term to the objective. WAE does not necessarily lead to a min-max game, uses a different penalty, and has a clear theoretical foundation.

Several works used reproducing kernels in context of GANs. Li et al. (2015); Dziugaite et al. (2015) use MMD with a fixed kernel $k$ to match $P_X$ and $P_G$ directly in the input space $\mathcal{X}$. These methods have been criticised to require larger mini-batches during training: estimating $\mathrm{MMD}_k(P_X, P_G)$ requires number of samples roughly proportional to the dimensionality of the input space $\mathcal{X}$ (Reddi et al., 2015) which is typically larger than $10^3$. Li et al. (2017) take a similar approach but further train $k$ adversarially so as to arrive at a meaningful loss function. WAE-MMD uses MMD to match $Q_Z$ to the prior $P_Z$ in the latent space $\mathcal{Z}$. Typically $\mathcal{Z}$ has no more than 100 dimensions and $P_Z$ is Gaussian, which allows us to use regular mini-batch sizes to accurately estimate MMD.

---

[3]Two popular choices are Gaussian and Bernoulli decoders $P_G(X|Z)$ leading to pixel-wise squared and cross-entropy losses respectively. In both cases the normalizing constants can be computed in closed form and don't depend on $Z$.

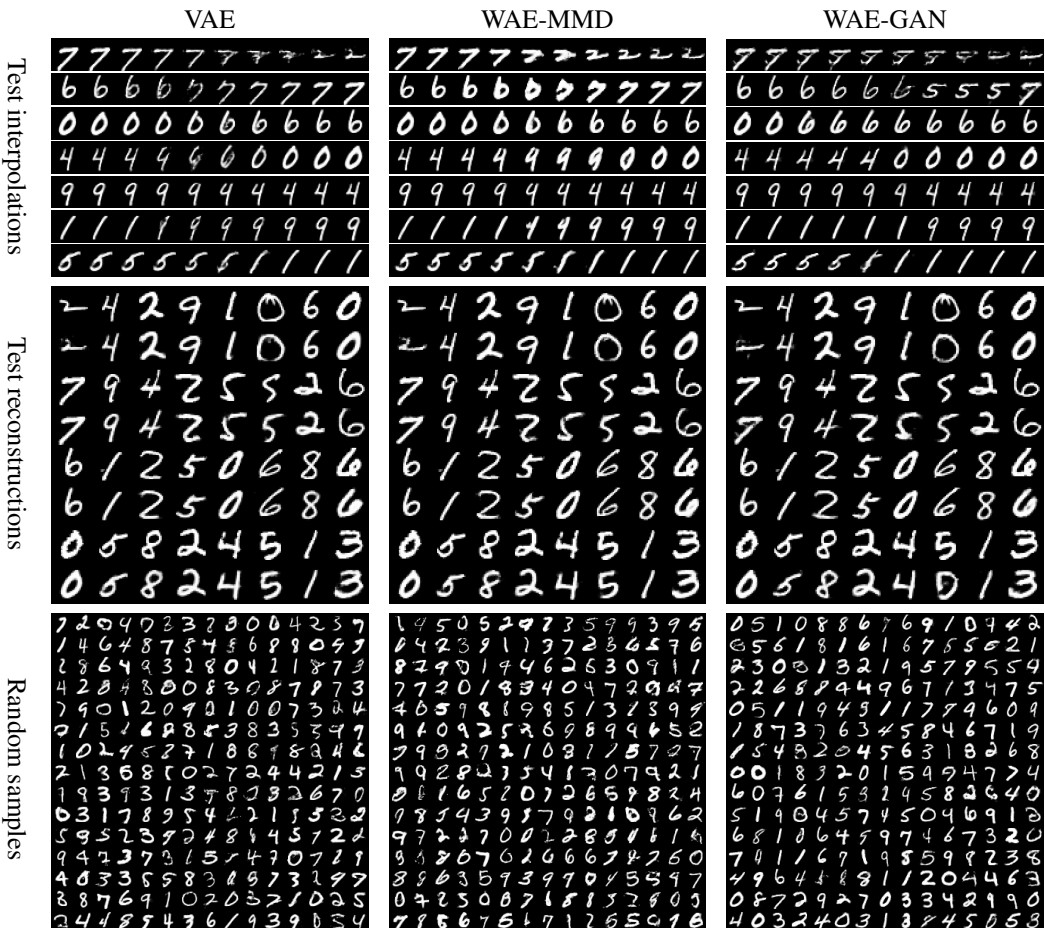

Figure 2: VAE (left column), WAE-MMD (middle column), and WAE-GAN (right column) trained on MNIST dataset. In "test reconstructions" odd rows correspond to the real test points.

## 4 EXPERIMENTS

In this section we empirically evaluate[4] the proposed WAE model. We would like to test if WAE can simultaneously achieve (i) accurate reconstructions of data points, (ii) reasonable geometry of the latent manifold, and (iii) random samples of good (visual) quality. Importantly, the model should generalize well: requirements (i) and (ii) should be met on both training and test data. We trained WAE-GAN and WAE-MMD (Algorithms 1 and 2) on two real-world datasets: MNIST (LeCun et al., 1998) consisting of 70k images and CelebA (Liu et al., 2015) containing roughly 203k images.

**Experimental setup** In all reported experiments we used Euclidian latent spaces $\mathcal{Z} = \mathcal{R}^{d_z}$ for various $d_z$ depending on the complexity of the dataset, isotropic Gaussian prior distributions $P_Z(Z) = \mathcal{N}(Z; \mathbf{0}, \sigma_z^2 \cdot \boldsymbol{I}_d)$ over $\mathcal{Z}$, and a squared cost function $c(x, y) = \|x - y\|_2^2$ for data points $x, y \in \mathcal{X} = \mathcal{R}^{d_x}$. We used *deterministic* encoder-decoder pairs, Adam (Kingma & Lei, 2014) with $\beta_1 = 0.5, \beta_2 = 0.999$, and convolutional deep neural network architectures for encoder mapping $\mu_\phi \colon \mathcal{X} \to \mathcal{Z}$ and decoder mapping $G_\theta \colon \mathcal{Z} \to \mathcal{X}$ similar to the DCGAN ones reported by Radford et al. (2016) with batch normalization (Ioffe & Szegedy, 2015). We tried various values of $\lambda$ and noticed that $\lambda = 10$ seems to work good across all datasets we considered.

Since we are using deterministic encoders, choosing $d_z$ larger than intrinsic dimensionality of the dataset would force the encoded distribution $Q_Z$ to live on a manifold in $\mathcal{Z}$. This would make matching $Q_Z$ to $P_Z$ impossible if $P_Z$ is Gaussian and may lead to numerical instabilities. We use $d_z = 8$ for MNIST and $d_z = 64$ for CelebA which seems to work reasonably well.

---

[4]The code is available at github.com/tolstikhin/wae.

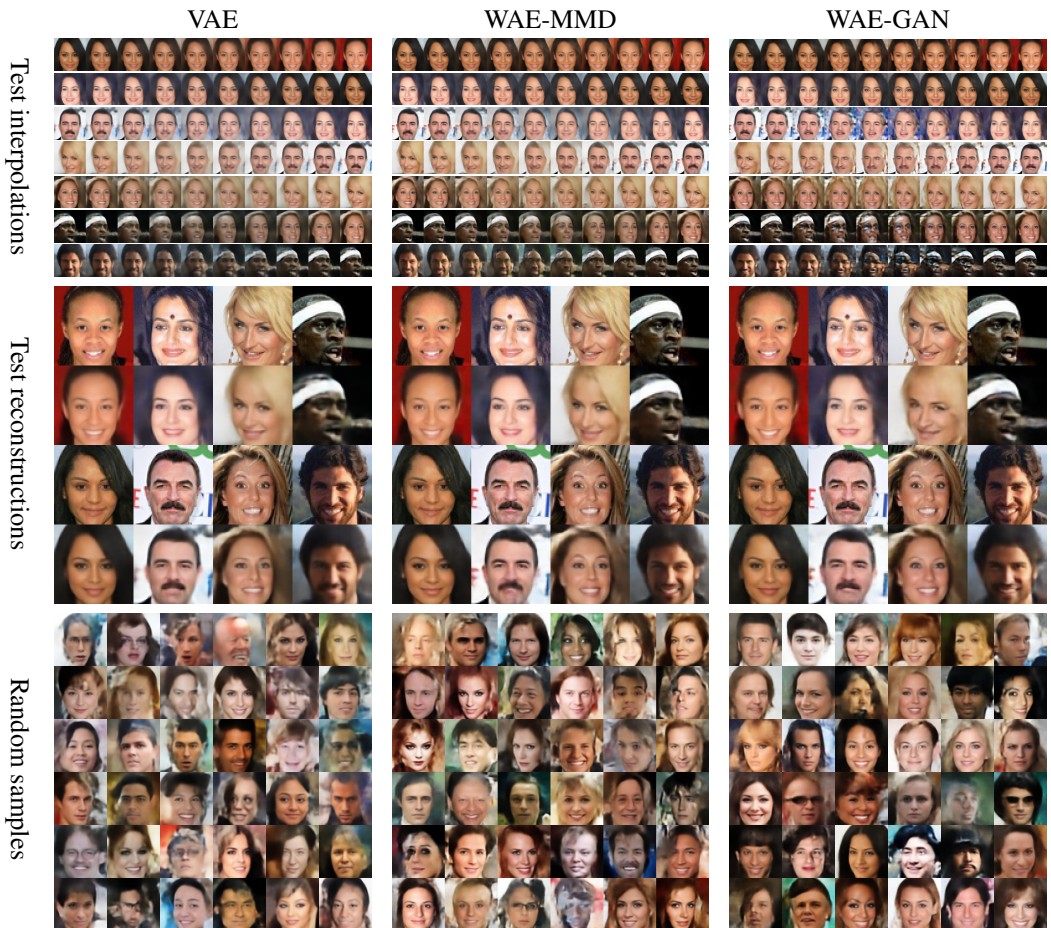

Figure 3: VAE (left column), WAE-MMD (middle column), and WAE-GAN (right column) trained on CelebA dataset. In "test reconstructions" odd rows correspond to the real test points.

We also report results of VAEs. VAEs used the same latent spaces as discussed above and standard Gaussian priors $P_Z = \mathcal{N}(\mathbf{0}, \mathbf{I}_d)$. We used Gaussian encoders $Q(Z|X) = \mathcal{N}\big(Z; \mu_\phi(X), \Sigma(X)\big)$ with mean $\mu_\phi$ and diagonal covariance $\Sigma$. For both MNIST and CelebA we used Bernoulli decoders parametrized by $G_\theta$. Functions $\mu_\phi$, $\Sigma$, and $G_\theta$ were parametrized by deep nets of the same architectures as used in WAE.

**WAE-GAN and WAE-MMD specifics** In WAE-GAN we used discriminator $D$ composed of several fully connected layers with ReLu. We tried WAE-MMD with the RBF kernel but observed that it fails to penalize the outliers of $Q_Z$ because of the quick tail decay. If the codes $\tilde{z} = \mu_\phi(x)$ for some of the training points $x \in \mathcal{X}$ end up far away from the support of $P_Z$ (which may happen in the early stages of training) the corresponding terms in the U-statistic $k(z, \tilde{z}) = e^{-\|\tilde{z}-z\|_2^2/\sigma_k^2}$ will quickly approach zero and provide no gradient for those outliers. This could be avoided by choosing the kernel bandwidth $\sigma_k^2$ in a data-dependent manner, however in this case per-minibatch U-statistic would not provide an unbiased estimate for the gradient. Instead, we used the *inverse multiquadratics* kernel $k(x, y) = C/(C + \|x - y\|_2^2)$ which is also characteristic and has much heavier tails. In all experiments we used $C = 2d_z\sigma_z^2$, which is the expected squared distance between two multivariate Gaussian vectors drawn from $P_Z$. This significantly improved the performance compared to the RBF kernel (even the one with $\sigma_k^2 = 2d_z\sigma_z^2$). Trained models are presented in Figures 2 and 3. Further details are presented in Supplementary C.

**Random samples** are generated by sampling $P_Z$ and decoding the resulting noise vectors $z$ into $G_\theta(z)$. As expected, in our experiments we observed that for both WAE-GAN and WAE-MMD the quality of samples strongly depends on how accurately $Q_Z$ matches $P_Z$. To

see this, notice that during training the decoder function $G_\theta$ is presented only with encoded versions $\mu_\phi(X)$ of the data points $X \sim P_X$. Indeed, the decoder is trained on samples from $Q_Z$ and thus there is no reason to expect good results when feeding it with samples from $P_Z$. In our experiments we noticed that even slight differences between $Q_Z$ and $P_Z$ may affect the quality of samples. In some cases WAE-GAN seems to lead to a better matching and generates better samples than WAE-MMD. However, due to adversarial training WAE-GAN is highly unstable, while WAE-MMD has a very stable training much like VAE.

In order to quantitatively assess the quality of the generated images, we use the *Fréchet Inception Distance* introduced by Heusel et al. (2017) and report the results on CelebA in Table 1. These results confirm that the sampled images from WAE are of better quality than from VAE, and WAE-GAN gets a slightly better score than WAE-MMD, which correlates with visual inspection of the images.

| Algorithm | FID |
|-----------|-----|
| VAE | 82 |
| WAE-MMD | 55 |
| WAE-GAN | 42 |

Table 1: FID scores for samples on CelebA (smaller is better).

**Test reconstructions and interpolations.** We take random points $x$ from the held out test set and report their auto-encoded versions $G_\theta(\mu_\phi(x))$. Next, pairs $(x, y)$ of different data points are sampled randomly from the held out test set and encoded: $z_x = \mu_\phi(x)$, $z_y = \mu_\phi(y)$. We *linearly* interpolate between $z_x$ and $z_y$ with equally-sized steps in the latent space and show decoded images.

## 5 CONCLUSION

Using the optimal transport cost, we have derived Wasserstein auto-encoders—a new family of algorithms for building generative models. We discussed their relations to other probabilistic modeling techniques. We conducted experiments using two particular implementations of the proposed method, showing that in comparison to VAEs, the images sampled from the trained WAE models are of better quality, without compromising the stability of training and the quality of reconstruction. Future work will include further exploration of the criteria for matching the encoded distribution $Q_Z$ to the prior distribution $P_Z$, assaying the possibility of adversarially training the cost function $c$ in the input space $\mathcal{X}$, and a theoretical analysis of the dual formulations for WAE-GAN and WAE-MMD.

### ACKNOWLEDGMENTS

The authors are thankful to Carl Johann Simon-Gabriel, Mateo Rojas-Carulla, Arthur Gretton, Paul Rubenstein, and Fei Sha for stimulating discussions.

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

## A   IMPLICIT GENERATIVE MODELS: A SHORT TOUR OF GANS AND VAES

Even though GANs and VAEs are quite different—both in terms of the conceptual frameworks and empirical performance—they share important features: (a) both can be trained by sampling from the model $P_G$ without knowing an analytical form of its density and (b) both can be scaled up with SGD. As a result, it becomes possible to use highly flexible *implicit* models $P_G$ defined by a two-step procedure, where first a code $Z$ is sampled from a fixed distribution $P_Z$ on a latent space $\mathcal{Z}$ and then $Z$ is mapped to the image $G(Z) \in \mathcal{X} = \mathcal{R}^d$ with a (possibly random) transformation $G \colon \mathcal{Z} \to \mathcal{X}$. This results in *latent variable models* $P_G$ of the form (3).

These models are indeed easy to sample and, provided $G$ can be differentiated analytically with respect to its parameters, $P_G$ can be trained with SGD. The field is growing rapidly and numerous variations of VAEs and GANs are available in the literature. Next we introduce and compare several of them.

The original **generative adversarial network** (GAN) Goodfellow et al. (2014) approach minimizes

$$D_{\mathrm{GAN}}(P_X, P_G) = \sup_{T \in \mathcal{T}} \mathbb{E}_{X \sim P_X}[\log T(X)] + \mathbb{E}_{Z \sim P_Z}\big[\log\big(1 - T(G(Z))\big)\big] \qquad (5)$$

with respect to a deterministic *decoder* $G \colon \mathcal{Z} \to \mathcal{X}$, where $\mathcal{T}$ is any non-parametric class of choice. It is known that $D_{\mathrm{GAN}}(P_X, P_G) \leq 2 \cdot D_{\mathrm{JS}}(P_X, P_G) - \log(4)$ and the inequality turns into identity in the *nonparametric limit*, that is when the class $\mathcal{T}$ becomes rich enough to represent *all* functions mapping $\mathcal{X}$ to $(0, 1)$. Hence, GANs are *minimizing a lower bound* on the JS-divergence. However, GANs are not only linked to the JS-divergence: the $f$-GAN approach Nowozin et al. (2016) showed that a slight modification $D_{\mathrm{f,GAN}}$ of the objective (5) allows to lower bound any desired $f$-divergence in a similar way. In practice, both decoder $G$ and *discriminator* $T$ are trained in alternating SGD steps. Stopping criteria as well as adequate evaluation of the trained GAN models remain open questions.

Recently, the authors of Arjovsky et al. (2017) argued that the 1-Wasserstein distance $W_1$, which is known to induce a much weaker topology than $D_{\mathrm{JS}}$, may be better suited for generative modeling. When $P_X$ and $P_G$ are supported on largely disjoint low-dimensional manifolds (which may be the case in applications), $D_{\mathrm{KL}}$, $D_{\mathrm{JS}}$, and other strong distances between $P_X$ and $P_G$ max out and no longer provide useful gradients for $P_G$. This "vanishing gradient" problem necessitates complicated scheduling between the $G/T$ updates. In contrast, $W_1$ is still sensible in these cases and provides stable gradients. The **Wasserstein GAN** (WGAN) minimizes

$$D_{\mathrm{WGAN}}(P_X, P_G) = \sup_{T \in \mathcal{W}} \mathbb{E}_{X \sim P_X}[T(X)] - \mathbb{E}_{Z \sim P_Z}\big[T(G(Z))\big],$$

where $\mathcal{W}$ is any subset of 1-Lipschitz functions on $\mathcal{X}$. It follows from (2) that $D_{\mathrm{WGAN}}(P_X, P_G) \leq W_1(P_X, P_G)$ and thus WGAN is *minimizing a lower bound* on the 1-Wasserstein distance.

**Variational auto-encoders** (VAE) Kingma & Welling (2014) utilize models $P_G$ of the form (3) and minimize

$$D_{\mathrm{VAE}}(P_X, P_G) = \inf_{Q(Z|X) \in \mathcal{Q}} \mathbb{E}_{P_X}\big[D_{\mathrm{KL}}\big(Q(Z|X), P_Z\big) - \mathbb{E}_{Q(Z|X)}[\log p_G(X|Z)]\big] \qquad (6)$$

with respect to a random *decoder* mapping $P_G(X|Z)$. The conditional distribution $P_G(X|Z)$ is often parametrized by a deep net $G$ and can have any form as long as its density $p_G(x|z)$ can be computed and differentiated with respect to the parameters of $G$. A typical choice is to use Gaussians $P_G(X|Z) = \mathcal{N}(X; G(Z), \sigma^2 \cdot I)$. If $\mathcal{Q}$ is the set of *all* conditional probability distributions $Q(Z|X)$, the objective of VAE coincides with the negative marginal log-likelihood $D_{\mathrm{VAE}}(P_X, P_G) = -\mathbb{E}_{P_X}[\log P_G(X)]$. However, in order to make the $D_{\mathrm{KL}}$ term of (6) tractable in closed form, the original implementation of VAE uses a standard normal $P_Z$ and restricts $\mathcal{Q}$ to a class of Gaussian distributions $Q(Z|X) = \mathcal{N}\big(Z; \mu(X), \Sigma(X)\big)$ with mean $\mu$ and diagonal covariance $\Sigma$ parametrized by deep nets. As a consequence, VAE is *minimizing an upper bound* on the negative log-likelihood or, equivalently, on the KL-divergence $D_{\mathrm{KL}}(P_X, P_G)$.

One possible way to reduce the gap between the true negative log-likelihood and the upper bound provided by $D_{\mathrm{VAE}}$ is to enlarge the class $\mathcal{Q}$. **Adversarial variational Bayes** (AVB) Mescheder et al. (2017) follows this argument by employing the idea of GANs. Given any point $x \in \mathcal{X}$, a noise $\epsilon \sim \mathcal{N}(0, 1)$, and any fixed transformation $e \colon \mathcal{X} \times \mathcal{R} \to \mathcal{Z}$, a random variable $e(x, \epsilon)$

implicitly defines one particular conditional distribution $Q_e(Z|X = x)$. AVB allows $\mathcal{Q}$ to contain all such distributions for different choices of $e$, replaces the intractable term $D_{\mathrm{KL}}\big(Q_e(Z|X), P_Z\big)$ in (6) by the adversarial approximation $D_{\mathrm{f,GAN}}$ corresponding to the KL-divergence, and proposes to minimize[5]

$$D_{\mathrm{AVB}}(P_X, P_G) = \inf_{Q_e(Z|X)\in\mathcal{Q}} \mathbb{E}_{P_X}\big[D_{\mathrm{f,GAN}}\big(Q_e(Z|X), P_Z\big) - \mathbb{E}_{Q_e(Z|X)}[\log p_G(X|Z)]\big]. \quad (7)$$

The $D_{\mathrm{KL}}$ term in (6) may be viewed as a regularizer. Indeed, VAE reduces to the classical unregularized auto-encoder if this term is dropped, minimizing the reconstruction cost of the encoder-decoder pair $Q(Z|X), P_G(X|Z)$. This often results in different training points being encoded into non-overlapping zones chaotically scattered all across the $\mathcal{Z}$ space with "holes" in between where the decoder mapping $P_G(X|Z)$ has never been trained. Overall, the encoder $Q(Z|X)$ trained in this way does not provide a useful representation and sampling from the latent space $\mathcal{Z}$ becomes hard Bengio et al. (2013).

**Adversarial auto-encoders** (AAE) Makhzani et al. (2016) replace the $D_{\mathrm{KL}}$ term in (6) with another regularizer:

$$D_{\mathrm{AAE}}(P_X, P_G) = \inf_{Q(Z|X)\in\mathcal{Q}} D_{\mathrm{GAN}}(Q_Z, P_Z) - \mathbb{E}_{P_X}\mathbb{E}_{Q(Z|X)}[\log p_G(X|Z)], \quad (8)$$

where $Q_Z$ is the marginal distribution of $Z$ when first $X$ is sampled from $P_X$ and then $Z$ is sampled from $Q(Z|X)$, also known as the *aggregated posterior* Makhzani et al. (2016). Similarly to AVB, there is no clear link to log-likelihood, as $D_{\mathrm{AAE}} \leq D_{\mathrm{AVB}}$. The authors of Makhzani et al. (2016) argue that matching $Q_Z$ to $P_Z$ in this way ensures that there are no "holes" left in the latent space $\mathcal{Z}$ and $P_G(X|Z)$ generates reasonable samples whenever $Z \sim P_Z$. They also report an equally good performance of different types of conditional distributions $Q(Z|X)$, including Gaussians as used in VAEs, implicit models $Q_e$ as used in AVB, and *deterministic* encoder mappings, i.e. $Q(Z|X) = \delta_{\mu(X)}$ with $\mu\colon \mathcal{X} \to \mathcal{Z}$.

# B  PROOF OF THEOREM 1 AND FURTHER DETAILS

We will consider certain sets of joint probability distributions of three random variables $(X, Y, Z) \in \mathcal{X} \times \mathcal{X} \times \mathcal{Z}$. The reader may wish to think of $X$ as true images, $Y$ as images sampled from the model, and $Z$ as latent codes. We denote by $P_{G,Z}(Y, Z)$ a joint distribution of a variable pair $(Y, Z)$, where $Z$ is first sampled from $P_Z$ and next $Y$ from $P_G(Y|Z)$. Note that $P_G$ defined in (3) and used throughout this work is the marginal distribution of $Y$ when $(Y, Z) \sim P_{G,Z}$.

In the optimal transport problem (1), we consider joint distributions $\Gamma(X, Y)$ which are called *couplings* between values of $X$ and $Y$. Because of the marginal constraint, we can write $\Gamma(X, Y) = \Gamma(Y|X)P_X(X)$ and we can consider $\Gamma(Y|X)$ as a non-deterministic mapping from $X$ to $Y$. Theorem 1. shows how to *factor* this mapping through $\mathcal{Z}$, i.e., decompose it into an encoding distribution $Q(Z|X)$ and the generating distribution $P_G(Y|Z)$.

As in Section 2.2, $\mathcal{P}(X \sim P_X, Y \sim P_G)$ denotes the set of all joint distributions of $(X, Y)$ with marginals $P_X, P_G$, and likewise for $\mathcal{P}(X \sim P_X, Z \sim P_Z)$. The set of all joint distributions of $(X, Y, Z)$ such that $X \sim P_X$, $(Y, Z) \sim P_{G,Z}$, and $(Y \perp\!\!\!\perp X)|Z$ will be denoted by $\mathcal{P}_{X,Y,Z}$. Finally, we denote by $\mathcal{P}_{X,Y}$ and $\mathcal{P}_{X,Z}$ the sets of marginals on $(X, Y)$ and $(X, Z)$ (respectively) induced by distributions in $\mathcal{P}_{X,Y,Z}$. Note that $\mathcal{P}(P_X, P_G)$, $\mathcal{P}_{X,Y,Z}$, and $\mathcal{P}_{X,Y}$ depend on the choice of conditional distributions $P_G(Y|Z)$, while $\mathcal{P}_{X,Z}$ does not. In fact, it is easy to check that $\mathcal{P}_{X,Z} = \mathcal{P}(X \sim P_X, Z \sim P_Z)$. From the definitions it is clear that $\mathcal{P}_{X,Y} \subseteq \mathcal{P}(P_X, P_G)$ and we immediately get the following upper bound:

$$W_c(P_X, P_G) \leq W_c^{\dagger}(P_X, P_G) := \inf_{P\in\mathcal{P}_{X,Y}} \mathbb{E}_{(X,Y)\sim P}\left[c(X, Y)\right]. \quad (9)$$

If $P_G(Y|Z)$ are Dirac measures (i.e., $Y = G(Z)$), it turns out that $\mathcal{P}_{X,Y} = \mathcal{P}(P_X, P_G)$:

---

[5]The authors of AVB Mescheder et al. (2017) note that using $f$-GAN as described above actually results in "unstable training". Instead, following the approach of Poole et al. (2016), they use a trained discriminator $T^*$ resulting from the $D_{\mathrm{GAN}}$ objective (5) to approximate the ratio of densities and then directly estimate the KL divergence $\int f\big(p(x)/q(x)\big)q(x)dx$.

**Lemma 1** $\mathcal{P}_{X,Y} \subseteq \mathcal{P}(P_X, P_G)$ *with identity if[6] $P_G(Y|Z = z)$ are Dirac for all $z \in \mathcal{Z}$.*

**Proof** The first assertion is obvious. To prove the identity, note that when $Y$ is a deterministic function of $Z$, for any $A$ in the sigma-algebra induced by $Y$ we have $\mathbb{E}\left[\mathbf{1}_{[Y \in A]}|X, Z\right] = \mathbb{E}\left[\mathbf{1}_{[Y \in A]}|Z\right]$. This implies $(Y \perp\!\!\!\perp X)|Z$ and concludes the proof. ■

We are now in place to prove Theorem 1. Lemma 1 obviously leads to

$$W_c(P_X, P_G) = W_c^\dagger(P_X, P_G).$$

The tower rule of expectation, and the conditional independence property of $\mathcal{P}_{X,Y,Z}$ implies

$$
\begin{aligned}
W_c^\dagger(P_X, P_G) &= \inf_{P \in \mathcal{P}_{X,Y,Z}} \mathbb{E}_{(X,Y,Z) \sim P}\left[c(X, Y)\right] \\
&= \inf_{P \in \mathcal{P}_{X,Y,Z}} \mathbb{E}_{P_Z} \mathbb{E}_{X \sim P(X|Z)} \mathbb{E}_{Y \sim P(Y|Z)}[c(X, Y)] \\
&= \inf_{P \in \mathcal{P}_{X,Y,Z}} \mathbb{E}_{P_Z} \mathbb{E}_{X \sim P(X|Z)}\left[c(X, G(Z))\right] \\
&= \inf_{P \in \mathcal{P}_{X,Z}} \mathbb{E}_{(X,Z) \sim P}\left[c(X, G(Z))\right].
\end{aligned}
$$

It remains to notice that $\mathcal{P}_{X,Z} = \mathcal{P}(X \sim P_X, Z \sim P_Z)$ as stated earlier.

### B.1 RANDOM DECODERS $P_G(Y|Z)$

If the decoders are non-deterministic, Lemma 1 provides only the inclusion of sets $\mathcal{P}_{X,Y} \subseteq \mathcal{P}(P_X, P_G)$ and we get the following upper bound on the OT:

**Corollary 1** *Let $\mathcal{X} = \mathcal{R}^d$ and assume the conditional distributions $P_G(Y|Z = z)$ have mean values $G(z) \in \mathcal{R}^d$ and marginal variances $\sigma_1^2, \ldots, \sigma_d^2 \geq 0$ for all $z \in \mathcal{Z}$, where $G \colon \mathcal{Z} \to \mathcal{X}$. Take $c(x, y) = \|x - y\|_2^2$. Then*

$$W_c(P_X, P_G) \leq W_c^\dagger(P_X, P_G) = \sum_{i=1}^d \sigma_i^2 + \inf_{P \in \mathcal{P}(X \sim P_X, Z \sim P_Z)} \mathbb{E}_{(X,Z) \sim P}\left[\|X - G(Z)\|^2\right]. \quad (10)$$

**Proof** First inequality follows from (9). For the identity we proceed similarly to the proof of Theorem 1 and write

$$W_c^\dagger(P_X, P_G) = \inf_{P \in \mathcal{P}_{X,Y,Z}} \mathbb{E}_{P_Z} \mathbb{E}_{X \sim P(X|Z)} \mathbb{E}_{Y \sim P(Y|Z)}\left[\|X - Y\|^2\right]. \quad (11)$$

Note that

$$
\begin{aligned}
\mathbb{E}_{Y \sim P(Y|Z)}\left[\|X - Y\|^2\right] &= \mathbb{E}_{Y \sim P(Y|Z)}\left[\|X - G(Z) + G(Z) - Y\|^2\right] \\
&= \|X - G(Z)\|^2 + \mathbb{E}_{Y \sim P(Y|Z)}\left[\langle X - G(Z), G(Z) - Y\rangle\right] + \mathbb{E}_{Y \sim P(Y|Z)}\|G(Z) - Y\|^2 \\
&= \|X - G(Z)\|^2 + \sum_{i=1}^d \sigma_i^2.
\end{aligned}
$$

Together with (11) and the fact that $\mathcal{P}_{X,Z} = \mathcal{P}(X \sim P_X, Z \sim P_Z)$ this concludes the proof. ■

## C FURTHER DETAILS ON EXPERIMENTS

### C.1 MNIST

We use mini-batches of size 100 and trained the models for 100 epochs. We used $\lambda = 10$ and $\sigma_z^2 = 1$. For the encoder-decoder pair we set $\alpha = 10^{-3}$ for Adam in the beginning and for the

---

[6]We conjecture that this is also a necessary condition. The necessity is not used in the paper.

adversary in WAE-GAN to $\alpha = 5 \times 10^{-4}$. After 30 epochs we decreased both by factor of 2, and after first 50 epochs further by factor of 5.

Both encoder and decoder used fully convolutional architectures with 4x4 convolutional filters.

Encoder architecture:

$$
\begin{aligned}
x \in \mathcal{R}^{28 \times 28} &\to \mathrm{Conv}_{128} \to \mathrm{BN} \to \mathrm{ReLU} \\
&\to \mathrm{Conv}_{256} \to \mathrm{BN} \to \mathrm{ReLU} \\
&\to \mathrm{Conv}_{512} \to \mathrm{BN} \to \mathrm{ReLU} \\
&\to \mathrm{Conv}_{1024} \to \mathrm{BN} \to \mathrm{ReLU} \to \mathrm{FC}_8
\end{aligned}
$$

Decoder architecture:

$$
\begin{aligned}
z \in \mathcal{R}^8 &\to \mathrm{FC}_{7 \times 7 \times 1024} \\
&\to \mathrm{FSConv}_{512} \to \mathrm{BN} \to \mathrm{ReLU} \\
&\to \mathrm{FSConv}_{256} \to \mathrm{BN} \to \mathrm{ReLU} \to \mathrm{FSConv}_1
\end{aligned}
$$

Adversary architecture for WAE-GAN:

$$
\begin{aligned}
z \in \mathcal{R}^8 &\to \mathrm{FC}_{512} \to \mathrm{ReLU} \\
&\to \mathrm{FC}_{512} \to \mathrm{ReLU} \\
&\to \mathrm{FC}_{512} \to \mathrm{ReLU} \\
&\to \mathrm{FC}_{512} \to \mathrm{ReLU} \to \mathrm{FC}_1
\end{aligned}
$$

Here $\mathrm{Conv}_k$ stands for a convolution with $k$ filters, $\mathrm{FSConv}_k$ for the fractional strided convolution with $k$ filters (first two of them were doubling the resolution, the third one kept it constant), BN for the batch normalization, ReLU for the rectified linear units, and $\mathrm{FC}_k$ for the fully connected layer mapping to $\mathcal{R}^k$. All the convolutions in the encoder used vertical and horizontal strides 2 and SAME padding.

Finally, we used two heuristics. First, we always pretrained separately the encoder for several mini-batch steps before the main training stage so that the sample mean and covariance of $Q_Z$ would try to match those of $P_Z$. Second, while training we were adding a pixel-wise Gaussian noise truncated at $0.01$ to all the images before feeding them to the encoder, which was meant to make the encoders random. We played with all possible ways of combining these two heuristics and noticed that together they result in *slightly* (almost negligibly) better results compared to using only one or none of them.

Our VAE model used cross-entropy loss (Bernoulli decoder) and otherwise same architectures and hyperparameters as listed above.

## C.2 CELEBA

We pre-processed CelebA images by first taking a 140x140 center crops and then resizing to the 64x64 resolution. We used mini-batches of size 100 and trained the models for various number of epochs (up to 250). All reported WAE models were trained for 55 epochs and VAE for 68 epochs. For WAE-MMD we used $\lambda = 100$ and for WAE-GAN $\lambda = 1$. Both used $\sigma_z^2 = 2$.

For WAE-MMD the learning rate of Adam was initially set to $\alpha = 10^{-3}$. For WAE-GAN the learning rate of Adam for the encoder-decoder pair was initially set to $\alpha = 3 \times 10^{-4}$ and for the adversary to $10^{-3}$. All learning rates were decreased by factor of 2 after 30 epochs, further by factor of 5 after 50 first epochs, and finally additional factor of 10 after 100 first epochs.

Both encoder and decoder used fully convolutional architectures with 5x5 convolutional filters.

Encoder architecture:

$$
\begin{aligned}
x \in \mathcal{R}^{64 \times 64 \times 3} &\to \mathrm{Conv}_{128} \to \mathrm{BN} \to \mathrm{ReLU} \\
&\to \mathrm{Conv}_{256} \to \mathrm{BN} \to \mathrm{ReLU} \\
&\to \mathrm{Conv}_{512} \to \mathrm{BN} \to \mathrm{ReLU} \\
&\to \mathrm{Conv}_{1024} \to \mathrm{BN} \to \mathrm{ReLU} \to \mathrm{FC}_{64}
\end{aligned}
$$

Decoder architecture:

$$z \in \mathcal{R}^{64} \rightarrow \text{FC}_{8 \times 8 \times 1024}$$
$$\rightarrow \text{FSConv}_{512} \rightarrow \text{BN} \rightarrow \text{ReLU}$$
$$\rightarrow \text{FSConv}_{256} \rightarrow \text{BN} \rightarrow \text{ReLU}$$
$$\rightarrow \text{FSConv}_{128} \rightarrow \text{BN} \rightarrow \text{ReLU} \rightarrow \text{FSConv}_1$$

Adversary architecture for WAE-GAN:

$$z \in \mathcal{R}^{64} \rightarrow \text{FC}_{512} \rightarrow \text{ReLU}$$
$$\rightarrow \text{FC}_{512} \rightarrow \text{ReLU}$$
$$\rightarrow \text{FC}_{512} \rightarrow \text{ReLU}$$
$$\rightarrow \text{FC}_{512} \rightarrow \text{ReLU} \rightarrow \text{FC}_1$$

For WAE-GAN we used a heuristic proposed in Supplementary IV of Mescheder et al. (2017). Notice that the theoretically optimal discriminator would result in $D^*(z) = \log p_Z(z) - \log q_Z(z)$, where $p_Z$ and $q_Z$ are densities of $P_Z$ and $Q_Z$ respectively. In our experiments we added the log prior $\log p_Z(z)$ explicitly to the adversary output as we know it analytically. This should hopefully make it easier for the adversary to learn the remaining $Q_Z$ density term.

Our VAE model used a cross-entropy reconstruction loss (Bernoulli decoder) and $\alpha = 10^{-4}$ as the initial Adam learning rate and the same decay schedule as explained above. Otherwise all the architectures and hyperparameters were as explained above.

