# OpenReview forum: "Wasserstein Auto-Encoders"
_ICLR.cc/2018/Conference — Accept (Oral)_

### Official Review · AnonReviewer2 · 2017-11-26
**This is a well-written paper which provides a useful generalisation of some existing methods for inferring generative models.**

**Rating:** 8
**Confidence:** 3

**Review:**

This paper satisfies the following necessary conditions for
acceptance. The writing is clear and I was able to understand the
presented method (and its motivation) despite not being too familiar
with the relevant literature. Explicitly writing the auto-encoder(s)
as pseudo-code algorithms was particular helpful. I found no technical
errors. The problem addressed is one worth solving - building a
generative model of observed data. There is some empirical testing
which show the presented method in a good light.

The authors are careful to relate the presented method with existing
ones, most notably VAE and AAE. I suppose one could argue that the
close connection to existing methods means that this paper is not
innovative enough. I think that would be unfair - most new methods
have close relations with existing ones - it is just that sometimes
the authors do not flag this up as they should.

WAE is a bit oversold. The authors state that WAE generates "samples
of better quality" (than VAE) without any condition being put on when
it does this. There is no proof that it is always better, and I can't
see how there could be. Any method of inferring a generative model
from data must make some 'inductive' assumptions. Surely one could
devise situations where VAE outperforms WAE. I think this issue should
have been examined in more depth.

I found no typo or grammatical errors which is unusual - good careful
job!

---

> ### Author Response · Authors · 2017-12-15
> **Answer to AnonReviewer2**
>
> We are pleased that the reviewer found the paper well written.
>
> We tried to be modest in our claims, in particular we never implied that WAEs produce better samples for *all data distributions*. As noticed by the reviewer this would be indeed impossible to prove, especially because the question of how to evaluate and compare sample qualities of unsupervised generative models is still open. We will double-check that there are no bold and unsupported statements in the final version of the paper.

---

### Official Review · AnonReviewer1 · 2017-11-29
**Excellent tutorial papers with novel contributions and convincing results**

**Rating:** 8
**Confidence:** 3

**Review:**

This very well written paper covers the span between W-GAN and VAE. For a reviewer who is not an expert in the domain, it reads very well, and would have been of tutorial quality if space had allowed for more detailed explanations. The appendix are very useful, and tutorial paper material (especially A).

While I am not sure description would be enough to reproduce and no code is provided, every aspect of the architecture, if not described, if referred as similar to some previous work. There are also some notation shortcuts (not explained) in the proof of theorems that can lead to initial confusion, but they turn out to be non-ambiguous. One that could be improved is P(P_X, P_G) where one loses the fact that the second random variable is Y.


This work contains plenty of novel material, which is clearly compared to previous work:
- The main consequence of the use of Wasserstein distance is the surprisingly simple and useful Theorem 1. I could not verify its novelty, but this seems to be a great contribution.
- Blending GAN and auto-encoders has been tried in the past, but the authors claim better theoretical foundations that lead to solutions that do not rquire min-max
- The use of MMD in the context of GANs has also been tried. The authors claim that their use in the latent space makes it more practival

The experiments are very convincing, both numerically and visually.

Source of confusion: in algorithm 1 and 2, \tilde{z} is "sampled" from Q_TH(Z|xi), some one is lead to believe that this is the sampling process as in VAEs, while in reality Q_TH(Z|xi) is deterministic in the experiments.

---

> ### Author Response · Authors · 2017-12-15
> **Answer to AnonReviewer1**
>
> We thank the reviewer for the positive feedback and the kind words regarding the overview part of the paper.
>
> We will make sure to make notations clearer and include all the details of architectures used in experiments in the updated version of the paper. Of course we will also open source the code.

---

### Official Review · AnonReviewer3 · 2017-12-02
**A well-written paper that generalizes Wasserstein distance to VAEs**

**Rating:** 8
**Confidence:** 4

**Review:**

This paper provides a reasonably comprehensive generalization to VAEs and Adversarial Auto-encoders through the lens of the Wasserstein metric. By posing the auto-encoder design as a dual formulation of optimal transport, the proposed work supports the use of both deterministic and random decoders under a common framework. In my opinion, this is one of the crucial contributions of this paper. While the existing properties of auto-encoders are preserved, stability characteristics of W-GANs are also observed in the proposed architecture. The results from MNIST and CelebA datasets look convincing, though could include additional evaluation to compare the adversarial loss with the straightforward MMD metric and potentially discuss their pros and cons. In some sense, given the challenges in evaluating and comparing closely related auto-encoder solutions, the authors could design demonstrative experiments for cases where Wassersterin distance helps and may be  its potential limitations.

The closest work to this paper is the adversarial variational bayes framework by Mescheder et.al. which also attempts at unifying VAEs and GANs. While the authors describe the conceptual differences and advantages over that approach, it will be beneficial to actually include some comparisons in the results section.

---

> ### Author Response · Authors · 2017-12-15
> **Answer to AnonReviewer3**
>
> We thank the reviewer for the positive feedback.
>
> Comparing properties of WAE-MMD and WAE-GAN is indeed an intriguing direction and we intend to look into the details in our future research. In this paper we only report initial empirical observations, which can be concluded by saying that WAE-MMD enjoys a stable training but does not match Pz and Qz perfectly, while the training of WAE-GAN is not so stable but leads to much better matches once succeeded.
>
> In this paper we decided that comparing to VAE was sufficient for our purposes: both VAE and AVB follow the same objective of maximizing the marginal log likelihood in contrast to the minimization of the optimal transport studied in our work. However, we do agree that in future it would be interesting to compute the FID scores of the AVB samples.

---

### Public Comment · (anonymous) · 2017-12-03
**Statement about the KL divergence term in VAEs**

You state in the paper that the variational auto-encoder objective is composed of reconstruction cost plus a KL divergence term that captures how distinct the image by the encoder of each training example is from the prior p(z), and then go on to say that this KL term is not guaranteeing that the overall encoded distribution matches the prior p(z).

However, as shown in the paper "ELBO surgery: yet another way to carve up the variational evidence lower bound" by Hoffman and Johnson, the KL term in the VAE objective can be decomposed into exactly this KL(q(z)||p(z)) between the average encoder distribution and the prior plus a mutual information term, and that the former is a heavy contributor towards the overall KL term. This means that VAE does indeed try to match the overall encoder distribution of q to the prior, but also includes a regularizing term that aims to minimize the mutual information between the hidden code z and the index of the observation x that encourages the VAE to have the encoder produce the same codes z for different observations.

In conclusion, it would be more accurate to state that in comparison to VAEs you simply exclude the mutual information regularisation term from the objective as formulated in the ELBO surgery paper.

---

> ### Author Response · Authors · 2018-02-17
> **WAEs drop the mutual information term in the VAE regularizer**
>
> Thank you for this comment. Indeed, this observation provides one more intuitively clear way to explain a difference between VAEs and WAEs. We will use your suggestion in the camera-ready version of the paper.

---

### Public Comment · ~Mathieu_Blondel1 · 2017-12-19
**Assumptions of Theorem 1 should be clarified**

Congratulations on this nice paper.

The ability to remove one of the two marginal constraints in Theorem 1 relies on the assumption that P_G(Y|Z=z) is a Dirac. I know that you stated in the intro that you focus on deterministic maps but it would be nice to repeat the assumptions made in Theorem 1.

---

> ### Author Response · Authors · 2017-12-19
> **Assumptions clarified**
>
> Dear Mathieu,
>
> thank you for the suggestion. We will update the paper accordingly.

---

### Public Comment · ~Min_Lin1 · 2018-01-19
**Possibility of a Markov Chain instead of Reconstruction.**

Thanks for the great work. It's nice to see there is theoretical support for the (auto-encoder + constraint on Z) objective.

It seems to me the expectation over X could not be moved out in theorem 1, as this breaks the independence of Z and X.
Consider the case Q(Z|X) is not deterministic, we can have a markov chain X_{t+1} ~ \int_Z [ P_G(X'|Z)Q(Z|X_t) ], which has a stationary distribution same as P_X.  The algorithm in this paper gives a special case where Q(Z|X) is deterministic and X_{t+1} = X_{t}.

In supplementary B, the case where the decoder is random is discussed. It would be nice to also discuss the cases where Q(Z|X) is random vs deterministic.

Do correct me if I'm wrong, thanks.

---

> ### Author Response · Authors · 2018-01-22
> **Regarding random / deterministic encoders in WAE**
>
> Thank you for the question.
>
> Unfortunately, we did not quite get the point of your Markov chain example. But we would like to make it clear that the paper does not assume anything specific about the encoder Q(Z|X). As long as the aggregated posterior Qz matches the prior Pz, the encoder can be either deterministic or random. The same holds true for the WAE algorithm. We will try to emphasize it better in the updated version of the paper.
>
> The decoder is indeed a different story: for Theorem 1 we need it to be deterministic, but a very similar result holds also for the random decoder (Supplementary B).

---

> > ### Public Comment · ~Min_Lin1 · 2018-01-22
> > **Clarification**
> >
> > Let me clarify the markov chain point.
> >
> > In the case Q(Z|X) is stochastic, the encode/decode chain X->Z->X' is stochastic. Namely, P(X'|X) is not a deterministic function, it is a distribution. A markov chain can be constructed if we sample X from P_X and use P(X'|X) as the transition probability.
> >
> > By optimizing the Wasserstein distance between P(X') and P_X, we hope to get the parameter such that P(X') == P_X. The reconstruction term in this paper requires that X' == X, which is stronger than P(X') == P_X.

---

### Public Comment · (anonymous) · 2018-02-25
**Limitation of theorem 1 should be stated clearer**

The key of the paper is Theorem 1. It says that we can estimate WS-distance between Px and Pg by optimizing over random encoders Q(Z|X). Implementation however assumes that Q(Z | X) is Gaussian with mean and covariance parametrized by deep neural nets. The discussion of Theorem 1 in the paper simply says: "Similarly to VAE, we propose to use deep neural networks to parametrize both encoders Q and decoders G." It should openly discuss the limitation of this parameterization because it significantly reduces the encoder space over which the model optimizes. Otherwise, readers and reviewers would be left with the impression that "wow, this approach avoids the min-max problem in GAN and formulate a min min problem, which is much easier to solve." In theory, it's true.  In practice, I suspect it's far from truth as the encoder space is significantly constrained and effectively forcing Q(Z) to match P(Z) is still an open problem.

---

> ### Author Response · Authors · 2018-02-25
> **There are no constraints on encoders**
>
> None of WAE implementations in our paper uses Gaussian encoders. Theorem 1 does not constrain encoders in any way and WAEs can be used with any form of random (or deterministic) encoders, including flexible implicit random encoders induced by generative architectures of GANs (and for instance used in [1]).
>
> [1] Mescheder, L. and Nowozin, S. and Geiger, A. Adversarial Variational Bayes: Unifying Variational Autoencoders and Generative Adversarial Networks. ICML 2017.

---

> > ### Public Comment · (anonymous) · 2018-02-25
> > **Is Q(Z|X) assumed to be Dirac?**
> >
> > Thank you for the clarification. I misinterpreted: "We also report results of VAEs. VAEs used the same latent spaces as discussed above and standard Gaussian priors P_Z = N(0; I_d). We used Gaussian encoders Q(Z|X) = N(Z;\mu_\phi(X); \Sigma(X)) with mean \mu_\phi and diagonal covariance \Sigma"
> >
> > You said: "Note that as opposed to VAEs, the WAE formulation allows for non-random encoders deterministically mapping inputs to their latent codes." Does that mean Q(Z|X) is assumed to be Dirac measure? If that's the case, Gamma(Y|X) is deterministic but the proof of Theorem 1 considers Gamma(Y|X) as non-deterministic mapping from X to Y. Am I wrong?

---

> > > ### Author Response · Authors · 2018-02-25
> > > **Encoders can indeed be Dirac**
> > >
> > > Yes, in Theorem 1 the encoder Q(Z|X) is allowed to be Dirac as long as it satisfies the constrain. Of course, it is not always the case that deterministic encoders can match the prior: consider the case when the intrinsic dimensionality of the data is less than the latent space dimensionality (as discussed on page 7 and in more details in [1]).
> > >
> > > Regarding Gamma(Y|X)---the conditional part of the coupling---we assume you are referring to the sentence "we can consider Gamma(Y|X) as a non-deterministic mapping from X to Y" appearing in the proof of Theorem 1 on page 13. The proof never argues that Gamma(Y|X) should be necessarily random or deterministic, and does not use any of these two assumptions.
> > >
> > > [1] Rubenstein, P., Scholkopf, B., Tolstikhin, I. On the Latent Space of Wasserstein Auto-Encoders. https://arxiv.org/pdf/1802.03761.pdf

---

> > > > ### Public Comment · (anonymous) · 2018-02-26
> > > > **Encoders must be random**
> > > >
> > > > The last line of the proof in page 14 says: "It remains to notice that \mathcal{P}_{X, Z} = \mathcal{P}(X ~ P_X, Z ~ P_Z) as stated earlier." That means the proof considers all possible coupling between X and Z and thus considers both random and deterministic encoders. So, the proof considers all form of Gamma(Y | X), whether random or deterministic.
> > > >
> > > > Theorem 1 is technically correct, but the framework's effectiveness in minimizing/estimating the WS-distance remains to be explored. We need to optimize over the set of all possible couplings between X and Z, which is impossible. Considering only Dirac or Gaussian encoders, however, might be too restrictive.

---

### Public Comment · ~Xiaojian_Ma1 · 2018-04-07
**A question to 'simply drop the mutual information term \mathbb{I}_Q(X,Z)'**

Thanks for your insightful work. I found a question during reading your paper. And I hope to get some guidance from you.

In Sec.3, you mentioned that compared to VAE, WAE just 'simply drop the mutual information term \mathbb{I}_Q(X,Z) in the VAE regularizer', where this entangled regularizer has been discussed in [1]. But It seems that the proof to this evident conclusion is not straight-forward. I've tried to prove the GAN-WAE version, by regarding the GAN objective as JSD(q(z|x) || p(z)), however, it seems that the entropy term still cannot be eliminated, which means that we may still suffer from this entangled term in the regularizer you proposed(at least the GAN-WAE version).

Is it possible for your to give some proof sketch on this conclusion? It will be better if the proof of both the GAN and MMD version can be provided. Thanks.

[1] Matthew D. Hoffman, Matthew J. Johnson, ELBO surgery: yet another way to carve up the variational evidence lower bound, 2016

---

> ### Author Response · Authors · 2018-04-09
> **On dropping the mutual information**
>
> Thank you for the question!
>
> Indeed, we were not completely clear in the text. The argument applies only to the version of WAE-GAN, where the KL divergence is used instead of the JS entropy (which can b estimated in a very similar way using the adversarial training).

---

### Decision · Program_Chairs · 2018-01-29
**ICLR 2018 Conference Acceptance Decision**

**Decision:**

Accept (Oral)

**Comment:**

This paper proposes a new generative model that has the stability of variational autoencoders (VAE) while producing better samples. The authors clearly compare their work to previous efforts that combine VAEs and Generative Adversarial Networks with similar goals.  Authors show that the proposed algorithm is a generalization of Adversarial Autoencoder (AAE) and minimizes Wasserstein distance between model and target distribution. The paper is well written with convincing results. Reviewers agree that the algorithm is novel and practical; and close connections of the algorithm to related approaches are clearly discussed with useful insights.  Overall, the paper is strong and I recommend acceptance.